# ALKBH5 Protects Against Hepatic Ischemia–Reperfusion Injury by Regulating YTHDF1-Mediated YAP Expression

**DOI:** 10.3390/ijms252111537

**Published:** 2024-10-27

**Authors:** Pixiao Wang, Mei Xiang, Ling Zhu, Rixin Zhang, Xiaolin Zheng, Zhi Zheng, Kai Li

**Affiliations:** 1Department of Hepatobiliary and Pancreatic Surgery, The Central Hospital of Wuhan, Tongji Medical College, Huazhong University of Science and Technology, Wuhan 430014, China; 2Department of Cardiology, The Central Hospital of Wuhan, Tongji Medical College, Huazhong University of Science and Technology, Wuhan 430014, China; xiangm@whu.edu.cn

**Keywords:** ALKBH5, ischemia–reperfusion injury, hepatocyte proliferation, apoptosis

## Abstract

Ischemia/reperfusion (I/R) injury with severe cell death is a major complication involved in liver transplantation and resection. The identification of key regulators improving hepatocyte activity may provide potential strategies to clinically resolve I/R-induced injury. N^6^-methyladenosine (m^6^A) RNA modification is essential for tissue homeostasis and pathogenesis. However, the potential involvement of m^6^A in the regulation of hepatocyte activity and liver injury has not been fully explored. In the present study, we found that hepatocyte AlkB homolog H5 (ALKBH5) levels were decreased both in vivo and in vitro I/R models. Hepatocyte-specific ALKBH5 overexpression effectively attenuated I/R-induced liver necrosis and improved cell proliferation in mice. Mechanistically, ALKBH5-mediated m^6^A demethylation improved the mRNA stability of YTH N^6^-methyladenosine RNA-binding protein 1 (YTHDF1), thereby increasing its expression, which consequently promoted the translation of Yes-associated protein (YAP). In conclusion, ALKBH5 is a regulator of hepatic I/R injury that improves hepatocyte repair and proliferation by maintaining YTHDF1 stability and YAP content. The ALKBH5–m^6^A–YTHDF1–YAP axis represents promising therapeutic targets for hepatic I/R injury to improve the prognosis of liver surgery.

## 1. Introduction

Hepatic ischemia/reperfusion (I/R) injury is a major complication in patients undergoing liver transplantation or hepatectomy [1,2]. The I/R injury process involves numerous complex events, including mitochondrial dysfunction, oxidative stress, and series of immune–inflammatory reactions. This complex network of events can culminate in liver injury after I/R or even induce liver failure [3]. Much has been learned about the inflammatory response induced by I/R [1]. However, less is known about the precise manner of the liver repair and recovery after I/R injury [4]. Meanwhile, no pharmacological approach for the treatment of I/R-triggered liver damage has been approved [3,5]. Therefore, further exploring endogenous regulatory mediators that function to effectively resolve injury and improve surgical prognosis is of great clinical importance.

N^6^-methyladenosine (m^6^A) modification as a way to regulate messenger RNAs (mRNA) is universal in mammals, which plays important roles in various biological processes. Typically, mRNA m^6^A modification is methylated by methyltransferase complexes (“writers”) and is removed by demethylase (“erasers”). In addition, the m^6^A marks of target mRNA can be recognized by m^6^A binding proteins (“readers”) to regulate its translation [6]. Human AlkB homolog H5 (ALKBH5) is a major m^6^A demethylase, which has attracted increasing attention among researchers, especially in the field of human liver diseases [7]. Previous studies have reported that ALKBH5 ameliorates liver fibrosis by suppressing the function of hepatic stellate cells and mitochondrial fission [8,9]. Additionally, Alkbh5 and FTO, two m^6^A demethylases, can coordinate to regulate cerebral ischemia–reperfusion injury by affecting neurons apoptosis, which involves the demethylation of the Bcl2 transcript and its protein expression levels [10]. Importantly, one recent study, by analyzing the gene expression from a dataset, revealed that several m^6^A regulators (including FTO, METTL3, and ALKBH5) are closely associated with hepatic I/R injury [11]. Another study showed that the expression of FTO is decreased during hepatic I/R injury. FTO can impair hepatic I/R injury via inhibiting Drp1-mediated mitochondrial fragmentation [12]. Nevertheless, to our knowledge, the precise role and potential mechanism of ALKBH5 in regulating liver I/R injury remains largely unknown. Additionally, previous reports have revealed that the Hippo pathway and its downstream effector YAP (Yes-associated protein) play important roles in controlling hepatocytes’ proliferation and apoptosis [13]. The activation of YAP attenuates hepatic damage and fibrosis in liver ischemia–reperfusion injury [14]. The canonical YAP activation is negatively regulated by Hippo upstream regulators, which dephosphorylates YAP and increases its nuclear translocation to regulate downstream gene expression. In addition, whether m^6^A modification is involved in YAP expression and its function remains unclear.

In the present study, we found that the expression of ALKBH5 in hepatocytes was suppressed during I/R injury. Conversely, the level of m^6^A-methylated RNA was elevated after I/R injury. Functionally, the adeno-associated virus-mediated liver-specific overexpression of ALKBH5 (AAV-ALKBH5) protected the liver against I/R injury. Mechanistically, ALKBH5-mediated m^6^A modification increased the expression of YTHDF1 via modulating the stability of mRNA, thus promoting the translation of YAP, a core regulator of cell proliferation. Collectively, ALKBH5 is a regulator of hepatic I/R injury that improves hepatocyte repair and proliferation by maintaining YTHDF1 stability and YAP contents.

## 2. Results

### 2.1. The Expression of m^6^A Demethylase ALKBH5 Is Decreased in Hepatocytes After Hepatic I/R Injury

To investigate the potential role of ALKBH5 in hepatic I/R injury, we established both an in vivo mouse model and an in vitro hepatocyte H/R model, as we have reported before [15]. The expression levels of ALKBH5 mRNA and protein were decreased in the livers of mice at 1, 6, and 24 h after the I/R injury procedure (Figure 1A,B), which was opposite to the gradual increase in m^6^A accumulation (Figure 1C). Interestingly, we observed that the ALKBH5 expression was up-regulated in hepatocytes that were treated with hypoxia for 1 h. Next, when the hepatocytes were subjected to H/R (1 h with hypoxia followed by 6 h of reperfusion), the ALKBH5 expression was dramatically decreased at both the mRNA and protein levels as compared to the control group (Figure 1D,E). Notably, the level of m^6^A methylated mRNA was down-regulated in the hepatocytes after the hypoxia treatment while being up-regulated after 6 h of reperfusion (Figure 1F). These results indicated that ALKBH5-mediated m^6^A demethylation was closely associated with the development of hepatic I/R injury.

### 2.2. ALKBH5 Overexpression Ameliorates I/R-Induced Hepatic Injury

To further explore the potential role of ALKBH5 in the progression of hepatic I/R injury, we used adeno-associated virus-carrying ALKBH5 (AAV8-Alkbh5) to specifically overexpress Alkbh5 in the mouse liver. Mice injected with AAV8-green fluorescent protein (GFP) were used as controls. A western blot analysis confirmed the ALKBH5 protein overexpression in the liver of the mice with the AAV8-Alkbh5 injection (Figure 2A). Compared to the control mice with GFP, the AAV8-Alkbh5 injected mice exhibited significantly improved liver function upon I/R injury, as indicated by their decreased serum hepatic transaminases (ALT and AST) levels (Figure 2B). Tissue damage was assessed using H&E staining (Figure 2C), and the Suzuki score of liver damage was consistently lower in the AAV8-Alkbh5 mice than in the GFP controls (Figure 2C). Apoptosis during liver I/R injury is directly involved in liver injury, which impairs liver function. The TUNEL staining of liver tissue after the I/R injury showed that the number of apoptotic cells were significantly lower in the AAV8-Alkbh5 group as compared to the GFP control (Figure 2D). On the contrary, the number of proliferating cellular nuclear antigen (PCNA) staining-positive cells was increased in the livers with ALKBH5 overexpression, indicating that the ALKBH5 promoted cell proliferation in the liver upon the I/R stimuli (Figure 2E). Meanwhile, ALKBH5 overexpression increased the protein expression of PCNA and Cyclin D1 (Figure 2F), indicating that more hepatocytes re-entered the cell cycle and initiated proliferation post-injury. In addition, the expression of anti-apoptotic protein B-cell lymphoma protein 2 (Bcl-2) was increased, while the expression of pro-apoptotic Bcl-2-associated X protein (Bax) was decreased in the liver of the AAV8-Alkbh5 mice after I/R injury, as compared to the GFP control group (Figure 2F). Next, we further examined several key proteins (YAP, Smad3, and Tgf-β1) that are necessary for liver regeneration after injury [16]. As shown in Figure 2G, Yes-associated protein (YAP) expression was upregulated in the liver with the ALKBH5 overexpression as compared to the GFP control group, while the phosphorylation of YAP at S127 (p-YAP) was not affected by ALKBH5 overexpression. Additionally, our results showed that the protein expression of Smad3 and Tgf-β1 were not affected by ALKBH5 overexpression (Figure 2G). Taken together, these results reveal that ALKBH5 protects against hepatic I/R injury, which might be attributed to its role in promoting proliferation and inhibiting cell death.

### 2.3. ALKBH5 Promotes the Protein Expression of YAP in Hepatocytes

We next examined whether the overexpression of ALKBH5 affected YAP expression in the cultured hepatocytes. Interestingly, the enforced expression of ALKBH5 had no effect on the expression of YAP mRNA (Figure 3A), while the protein expression of YAP was consistently increased by the ALKBH5 overexpression (Figure 3B). Additionally, both the mRNA and protein expression of the YAP target genes (BIRC5, SPP1, and MYC) were increased by the ALKBH5 overexpression in the cultured L02 cells (Figure 3A,B). Additionally, we examined the expression levels of several signaling molecules implicated in the regulation of hepatocyte proliferation and liver injury [17]. In contrast, the expression of the core components of the Wnt/β-catenin (β-catenin and Axin1) and Notch pathways (Notch1) were not affected by the ALKBH5 overexpression in the H/R-treated L02 cells, as indicated by the qPCR (Figure 3A) and western blot (Figure 3B) analysis. Additionally, our results showed that the ALKBH5 protein was only detectable in the nuclear fractions, enabling the m6A modification of pre-mRNA, which was consistent with previous studies [18] (Figure 3C). Notably, the ALKBH5 overexpression increased the YAP contents in both the nucleus and cytoplasm (Figure 3C). By contrast, the ALKBH5 overexpression did not affect the protein expression of β-catenin, neither in the nucleus nor in the cytoplasm (Figure 3C). These results clearly indicate that YAP expression is controlled by ALKBH5 in hepatocytes upon H/R stimuli.

### 2.4. ALKBH5 Regulates YTHDF1 Expression via m^6^A Demethylation to Promote the Translation of YAP

To determine the molecular mechanism of ALKBH5 in regulating the expression of YAP, we evaluated the m^6^A modification of YAP in the H/R-treated L02 cells with ALKBH5 overexpression. Intriguingly, we observed the evident m^6^A methylation of YAP mRNA in the control group, which was not affected by the ALKBH5 overexpression (Figure 4A). A previous study demonstrated that YTH N^6^-methyladenosine RNA-binding protein 1 (YTHDF1), the m6A reader, is a key downstream factor of ALKBH5 to mediate YAP expression [19]. Consistently, our further m^6^A-specific qPCR analysis of YTHDF1 showed that the m^6^A methylation levels of YTHDF1 were significantly decreased in the ALKBH5 overexpression group (Figure 4B). Moreover, ALKBH5-knockdown (si-ALKBH5) (Appendix A) reduced the expression of YTHDF1 mRNA (Figure 4C) and protein (Figure 4D). As m^6^A modification regulates gene expression mainly by affecting the stability of mRNA, we measured the half-lives of YTHDF1 mRNAs in the cultured hepatocytes after transfection with an ALKBH5-expressing plasmid or targeting siRNA. As expected, the half-life of the YTHDF1 mRNA decreased remarkably in response to the ALKBH5-knockdown while being increased by the ALKBH5 overexpression (Figure 4E). Meanwhile, the protein expression of YTHDF1 was consistently decreased by the ALKBH5-knockdown in the cultured hepatocytes when treated with Actinomycin D (Figure 4F). Moreover, an RIP-qPCR analysis revealed the interaction of YAP with YTHDF1 in the H/R-treated hepatocytes (Figure 4G). These results indicate that ALKBH5 regulates the stability of YTHDF1 via m^6^A demethylation.

### 2.5. ALKBH5 Promotes Hepatocyte Recovery from H/R-Induced Damage by Regulating YTHDF1-YAP Expression

Next, we studied the role of YTHDF1 in regulating hepatocyte recovery from H/R-induced damage. The YTHDF1 siRNAs (si-YTHDF1) were transfected to the cultured hepatocytes, and the constructs with the most efficient knockdown expression were used in the following study (Appendix A). As demonstrated by flow cytometry assays with CFSE staining, YTHDF1-knockdown (si-YTHDF1) noticeably inhibited the proliferation of the hepatocytes as compared to the cells transfected with the negative control siRNA (si-NC), even in the presence of ALKBH5 overexpression (OE-ALKBH5) (Figure 5A). Meanwhile, compared to the si-NC control group, the si-YTHDF1-infected cells exhibited a significantly elevated apoptosis rate in response to the H/R injury, as revealed by the flow cytometry assays (Figure 5B). In addition, the si-YTHDF1 significantly decreased cell viability after the stimulation with H/R, as compared to the si-NC control group (Figure 5C). Next, we examined the expression of specific markers related to cell death or cell proliferation using a qPCR (Figure 5D) and western blot (Figure 5E) analysis. To be specific, Ki-67 is highly expressed in proliferating cells (maximally in the G2 and early M phase), is rapidly degraded during anaphase and telophase, and is rarely expressed in the G0 phase [20]. PCNA marks proliferating cells and is most highly expressed in S-phase cells [21]. We observed a decreased expression in cell proliferation markers (PCNA and Ki-67) and an increased expression in cell death markers (Cytochrome c, BAX, and cleaved caspase-3) in the si-YTHDF1 group relative to the si-NC control group, both at the mRNA (Figure 5D) and protein levels (Figure 5E). Notably, our data showed that the YTHDF1-knockdown obviously inhibited the YAP expression in comparison to the si-NC control group, even in the presence of ALKBH5 overexpression (si-YTHDF1 + OE-ALKBH5) (Figure 5E). Collectively, these results demonstrate that YTHDF1 is a key mediator that regulates YAP expression, which promotes hepatocyte recovery from H/R-induced damage.

### 2.6. YAP Depletion Exacerbates ALKBH5-Overexpression Potentiated Recovery of Hepatocyte from H/R Induced Damage

To validate that the protective effect of ALKBH5 on hepatocytes upon H/R injury is dependent on YAP expression, we co-transfected hepatocytes with YAP siRNA (Si-YAP) and ALKBH5 overexpression plasmids (OE-ALKBH5). A qPCR analysis of the YAP expression confirmed the knockdown efficiency of Si-YAP (Appendix A). Next, flow cytometry assays with CFSE staining showed that YAP-knockdown inhibited the proliferation of the hepatocytes after the stimulation with H/R, regardless of ALKBH5 overexpression (Figure 6A). Moreover, the flow cytometry assays also showed that the YAP-knockdown counteracted the antiapoptotic effect of the ALKBH5 overexpression on the hepatocytes challenged with H/R (Figure 6B). As demonstrated by the CCK8 assay, the YAP-knockdown (Si-YAP) significantly decreased cell viability in comparison to the si-NC control group (Figure 6C). Meanwhile, OE-ALKBH5 significantly increased cell viability, which was attenuated by the YAP-knockdown (Figure 6C). Furthermore, the qPCR (Figure 6D) and western blot (Figure 6E) results showed decreased levels of cell proliferation markers (PCNA and Ki-67), along with elevated levels of cell death markers (Cytochrome c, Bax, and cleaved caspase-3) in the si-YAP group, as compared to the si-NC control group (Figure 6D,E). Consistently, the regulatory effects of the ALKBH5 on the expression levels of these downstream factors was also attenuated by the YAP-knockdown (Si-YAP + OE-ALKBH5), regardless of the YTHDF1 levels (Figure 6D,E). Subsequently, an immunostaining analysis further confirmed the influence of ALKBH5-YAP on the expression levels of Ki-67 and Bax in the hepatocytes under the H/R conditions (Figure 6F). In summary, these results suggest that YAP expression is responsible for ALKBH5 function in hepatocyte proliferation and apoptosis in response to H/R damage.

## 3. Discussion

In the present study, we uncovered the function of ALKBH5 as a protector against I/R-induced liver injury. We found the decreased expression of Alkbh5 in hepatocytes along with the severe damage of I/R-induced liver injury. The AAV-mediated overexpression of ALKBH5 increased the ability of hepatocyte proliferation, reduced cell death, and restored hepatic function after I/R injury. Mechanistically, under normal conditions, m^6^A “eraser” ALKBH5 downregulates the methylation of m^6^A “reader” YTHDF1 to promote the stability of YTHDF1 mRNA, and then YTHDF1 promotes the efficient translation of YAP. On the contrary, upon I/R stimuli, decreased ALKBH5 in the hepatocyte is accompanied with decreased YTHDF1, resulting in a low level of YAP content (Figure 7). Collectively, our results revealed the critical role of the ALKBH5–m^6^A–YTHDF1–YAP axis in hepatic I/R injury.

The liver is a unique organ in terms of its potential regenerative capacity. Specifically, hepatocytes account for the majority of the liver mass and provide most of the hepatic functions related to body homeostasis. Hepatocytes rarely proliferate in their quiescent phase; however, once the liver is suffering from some special stimulus, hepatocytes can gain the potential of proliferation for maintaining organ function and sometimes restore its original size [17]. Therefore, strategies to fine-tune the condition of hepatocytes, such as inhibiting hepatocyte damage and cell death, while promoting the re-activation of hepatocyte proliferation after injury are particularly important [4]. Our study is the first to demonstrate the protective role of m^6^A demethylase ALKBH5 in the context of liver I/R injury. Notably, recent studies have uncovered the regulatory roles of ALKBH5 in other organs under similar circumstances. ALKBH5 mitigates the I/R injury-induced apoptosis of cardiomyocytes through promoting the stability of SIRT1 [22]. Similarly, ALKBH5 suppresses the m^6^A methylation of MG53 and inhibits MG53 degradation to inhibit cardiomyocyte apoptosis during the myocardial infarction process [23]. Additionally, ALKBH5 protects against ischemic stroke by reducing neuronal apoptosis [24]. Interestingly, but unexpectedly, ALKBH5 exacerbates I/R-induced renal injury [25], which indicates that multiple functions of ALKBH5 rely on different downstream targets in a cell type-dependent manner. Beyond these impressive findings, our results showed that the enforced expression of ALKBH5 inhibits cell death and promotes hepatocyte proliferation by enhancing YAP expression.

In the mammalian liver, YAP has been identified as a core factor in controlling cell fate and homeostasis maintenance [26]. On one hand, the expression levels of YAP in hepatocytes are critical factors in liver homeostasis [27]. YAP-null hepatocytes are more prone to apoptosis, and the overexpression of YAP induces liver enlargement in mice [28]. The loss of YAP in the liver leads to defects in both hepatocyte survival and biliary epithelial cell development [29]. Furthermore, the hepatocyte-specific deletion of YAP displayed major delays in liver regeneration post-hepatectomy due to decreased hepatocytes [30], whereas the overexpression of YAP can induce cell hypertrophy and proliferation in several different cell lines [31]. On the other hand, YAP activation has also been demonstrated to be essential for liver repair and regeneration [32]. In particular, Miyamura et al. [33] revealed that YAP could exert dual action in controlling the cell fate of hepatocytes in response to injury. YAP activation in damaged hepatocytes specifically promotes their elimination, while YAP activation in undamaged hepatocytes leads to proliferation [33]. Additionally, mice with liver-specific YAP activation were protected against I/R injury and fibrosis in a Nrf2-dependent manner [14]. Consistently with previous reports, we found that interfering with YAP expression induced obvious cell death and decreased hepatocyte proliferation. Furthermore, the protective effects of ALKBH5 were abolished when YAP expression was inhibited. Thus, the present study enhances our understanding of the important role the ALKBH5–YAP axis in promoting the recovery of the liver from I/R injury and sustaining homeostasis.

The Hippo–YAP signaling pathway could exert a key function in many aspects of liver biology, such as liver development and regeneration, hepatocyte proliferation, and homeostatic function recovery [26]. YAP is ordinarily inactive in cytoplasm due to phosphorylation by the Hippo kinases that maintain quiescence in the liver. Upon specific stimuli or injuries, dephosphorylated YAP shuttles from the cytoplasm into the nucleus and usually coactivates a transcriptional enhancer factor domain family member (TEAD), which then regulates a number of genes involved in hepatocyte growth, proliferation, and dedifferentiation [34]. The canonical Hippo signaling in regulating YAP activity has been well described previously [26]. However, the fine-tuned regulation of YAP expression still remains elusive. Our results showed that the protein level of YAP was significantly increased by ALKBH5 overexpression. Interestingly, our further study demonstrated that the ALKBH5 expression status had no effect on the m^6^A methylation of the YAP mRNA. Instead, the ALKBH5-mediated m^6^A demethylation enhanced the YTHDF1 mRNA stability and protein expression, thereby indirectly promoting the YAP translation. Most notably, one separate study previously revealed the consistent working mode of ALKBH5 in regulating YAP expression to promote cardiomyocyte proliferation. The study consistently demonstrated that the ALKBH5 promoted the stability of the YTHDF1 mRNA and increased the translation of YAP [19]. YTHDF1 has a well-defined role in recognizing the m^6^A-modified sites of target mRNA and promoting its translation [35]. YTHDF1-regulated STAT5 translation mediates the protective role of ALKBH5 in stroke [24]. Wang et al. [8] reported that the loss of ALKBH5 increases Drp1 protein expression levels depending on YTHDF1-mediated translation. To be specific, ALKBH5 downregulation contributes to more m^6^A modification of Drp1 mRNA, and then the m^6^A reader YTHDF1 promotes Drp1 translation by recognizing its m^6^A-modified sites. Collectively, these results reveal that YTHDF1 acts as an ALKBH5-binding protein, and YTHDF1 could be a bridge factor to mediate the expression levels of target proteins. All these present findings may help deepen our understanding of the molecular mechanism of m^6^A methylation. However, the exact mechanisms of the relationship between the family members of m^6^A regulators in normal and abnormal biological processes need to be further studied.

In summary, to the best of our knowledge, this study is the first to demonstrate that hepatocyte ALKBH5 is a protective factor against hepatic I/R injury. Specifically, the ALKBH5-mediated m^6^A demethylation upregulated the YTHDF1 expression levels, thereby promoting the translation of YAP, which is important for sustaining hepatocellular homeostasis. Targeting the ALKBH5–m^6^A–YTHDF1–YAP regulatory axis is thus a strategy to interfere with hepatic I/R injury and promote the liver’s recovery from injury.

## 4. Materials and Methods

### 4.1. Animals

Male C57BL/6 mice (6–8 weeks) were used as a wild type and background mice in this study. All animals were housed in a specific pathogen-free environment (controlled temperature, 23 ± 2 °C; 12 h light/dark photocycle). Food and water were available ad libitum. To specifically overexpress ALKBH5 in the liver, an adeno-associated virus serotype 8 (AAV8) carrying a full-length ALKBH5 coding sequence (NC_000017.11) under the trigger of the albumin promoter (AAV- ALKBH5) was used (*n* = 6). The AAV8 vector carrying GFP scramble (AAV-GFP) was used as a negative control (*n* = 6). AAV expression vectors were constructed and verified by the Wuhan Institute of Advanced Technology, Chinese Academy of Science. The expression vectors were injected into the mice through the tail vein (2 × 10^12^ viral genomes per mouse in 100 μL saline). The I/R injury model was performed 4 weeks after the virus injection. All animal experiments were approved by the Animal Ethics Committee of Wuhan Bestcell Model Biological Center (BSMS 2023-10-19A, date of approval: 19 October 2023).

### 4.2. Mouse Hepatic Ischemia/Reperfusion (I/R) Injury Model

We established a 70% warm I/R injury model in the mouse liver, as previously described [15]. Briefly, after anesthetization, the abdominal cavity of mice was opened, and then the blood vessels supplying the left and median lobes (ischemic lobes) of the liver were separated and clamped with vascular clips. After 1 h of ischemia, the reperfusion of the ischemic lobes was initiated by removing the vascular clips. Mice in the sham group received the same surgical procedure without vasculature occlusion. After reperfusion for indicated time points, liver and blood samples of the mice were collected for subsequent analysis.

### 4.3. Mouse Liver Function Measurement

To assess the severity of the liver injury, the serum levels of alanine aminotransferase (ALT) and aspartate aminotransferase (AST) were measured using a fully automatic biochemical analyzer (ADVIA 2400; Siemens, New York, NY, USA) according to the manufacturer’s instructions [36].

### 4.4. Histological Analysis

Liver paraffin sections (5 μm) were stained with hematoxylin and eosin (H&E). Then, the stained sections were photographed and the necrotic area was quantitatively assessed. The severity of the liver injury was graded by analyzing the necrotic area of the picture using Suzuki’s criteria: 0, no liver necrosis; 1, single cell necrosis; 2, up to 30% lobular necrosis; 3, up to 60% lobular necrosis; and 4, more than 60% lobular necrosis.

### 4.5. Immunofluorescence and TUNEL Staining

Paraffin-embedded liver sections were stained by immunofluorescence, as previously described [15]. Briefly, liver sections were firstly incubated with 10% bovine serum albumin for 1 h at 37 °C. Then, the sections were incubated with the indicated primary antibodies at 4 °C overnight, followed by an incubation with goat anti-rabbit IgG (#A-11011, Thermo, Waltham, MA, USA) as secondary antibody for 1 h. Mouse anti-PCNA (#2586, CST, Danvers, MA, USA) was used as primary antibody to visualize the proliferation cells. TUNEL staining was performed to evaluate the apoptotic cells according to the manufacturer’s instructions (Apoptosis Fluorescein Detection Kit, #S7111, Millipore, Burlington, MA, USA). The numbers of positive cells in liver sections were blindly evaluated by counting the labelled cells in 10 high-power fields (HPFs)/sections (40×). The nuclei were stained with DAPI (#S36939, Invitrogen, Waltham, MA, USA).

### 4.6. Cell Culture and Hepatocyte Hypoxia/Reoxygenation (H/R) Model

Human hepatocyte cell line L02 cells were purchased from the Type Culture Collection of the Chinese Academy of Sciences (Shanghai, China) and maintained in DMEM containing 10% fetal bovine serum (FBS, catalog no. 10099141; Thermo, Waltham, MA, USA) in an incubator chamber under normal conditions (at 37 °C with 5% CO_2_). To establish an H/R model in vitro, the cells were challenged with hypoxia for 1 h (1% O_2_, 5% CO_2_ and 94% N_2_); meanwhile, the medium of the cells was replaced with a serum-free DMEM/F12 medium. Subsequently, for the reoxygenation group, the medium of the cells was aspirated and replaced with a fresh and warmed normal maintenance medium, and the plates were returned to the incubator chamber under normal conditions for the indicated times. The medium and cells were collected for further analysis.

### 4.7. qPCR Assay

The qPCR assay was performed as previously reported [37]. In brief, the total RNA was extracted from the frozen liver tissue or cultured cell samples using a TRIzol reagent (15596-026, Invitrogen, Waltham, MA, USA). A total of 2 µg RNA was reverse-transcribed into cDNA using the Transcriptor First Strand cDNA Synthesis Kit (#04896866001, Roche, Basel, Switzerland). Quantitative real-time PCR was performed using SYBR green (#04887352001, Roche) to determine the expression levels of the target genes. The mRNA expression levels of the target genes were normalized to GAPDH. The primer pairs used in this study are listed in Appendix A.

### 4.8. Western Blot

A western blot assay was used to determine the target protein expression in the liver tissue or cell samples, as previously described [37]. Briefly, the extracted protein samples (50 μg) were separated on a 10% SDS–PAGE gel and transferred to a PVDF membrane. After blocking with 5% skim milk, the membrane was incubated with primary antibodies at 4 °C overnight, followed by incubation with the corresponding secondary antibodies for 1 h at room temperature. The expression levels of the target proteins were normalized to GAPDH or β-actin. The antibodies used in the present study are listed in Appendix A.

### 4.9. m^6^A Measurements

The level of global m^6^A in the total RNA was measured using an m^6^A RNA Methylation Quantification Kit (#ab185912, Abcam, Cambridge, UK) following the manufacture’s protocol. Briefly, messenger RNA was isolated from the cells and coated on assay wells for 90 min. Each well was washed, and the captured antibody, detection antibody, and enhancer antibody were added. Color developing solution was then added. The m6A levels were quantified using a microplate reader at a wavelength of 450 nm.

### 4.10. RNA Immunoprecipitation (RIP)-qPCR

The RIP experiment was performed according to a previous report [19]. Antibodies against YTHDF1 or a negative IgG control was used. For the m^6^A-specific RIP-qPCR, a Magna RIP kit (Millipore, Cat# 17-700) and antibody against m6A were used to examine the m^6^A modification on genes. In brief, the cells were lysed in RIP lysis buffer, and then the samples mixed with the indicated antibody were immunoprecipitated using A/G magnetic beads. After rotation at 4 °C overnight, the beads were washed to remove the unbound substance. Then, RNA was extracted and purified according to the manufacturer’s instructions. Finally, the purified RNA was analyzed by qPCR.

### 4.11. Cell Viability Assay

The cell viability was detected using the Cell Counting Kit-8 (CCK-8, Solarbio, Beijing, China), following the manufacture’s protocol. After the H/R treatment for the indicated time, 10% CCK8 reagent was added to the cultured cells and incubated for 60 min. The optical density (OD) value was measured using a microplate reader at the wavelength of 450 nm (ST-360, Technogenetics S.P.A., Shanghai, China).

### 4.12. Carboxyfluorescein Diacetate, Succinimidyl Ester (CFSE) Proliferation Assay

The CFSE-based proliferation assay was performed according to a previous report [38]. In brief, the cultured cells were labelled with CFSE (Cat# C1031, Beyotime, Shanghai, China) for 10 min at 37 °C. After the H/R treatment for the indicated time, the cells were washed twice with warm PBS, detached, and counted. Then, the cell proliferation was immediately analyzed by flow cytometry.

### 4.13. Flow Cytometry

Flow cytometry analysis was performed according to our previous report [36]. Briefly, after the H/R treatment for the indicated time, the cells were stained with 10 μL FITC-labeled Annexin V (GeMExo Biotech Corp, Wuhan, China) and 5 μL propidium iodide (PI; GeMExo Biotech Corp, Wuhan, China) at room temperature and in the dark for 15 min. The stained cells were analyzed using a flow cytometer (CytoFLEX S, Beckman Coulter, Brea, CA, USA).

### 4.14. Plasmid Transfection

The plasmids for regulating gene expression used in this study were constructed by Jiangsu Genecefe Biotechnology Co. (Wuxi, China), including siRNA targeting YAP, ALKBH5 and YTHDF1 and ALKBH5-carrying plasmid. The cells were transfected with 500 ng plasmid using Lipofectamine Transfection Reagent (Lipofectamine 2000 Reagent, Thermo, Waltham, MA, USA) according to the manufacturer’s protocols. The primer sequences of the siRNAs or ALKBH5 overexpression constructs are listed in Appendix A.

### 4.15. Statistical Analysis

All the data are expressed as the mean ± SD. A two-tailed Student’s *t*-test (unpaired) was used for comparisons between the two groups. For multiple comparisons, a one-way analysis of variance (ANOVA) followed by a least squares difference (LSD) test (assuming equal variances) or Tamhane’s T2 test (without the assumption of equal variances) were performed using GraphPad Prism software version 8. Differences were considered significant when *p* < 0.05.

## Figures and Tables

**Figure 1 ijms-25-11537-f001:**
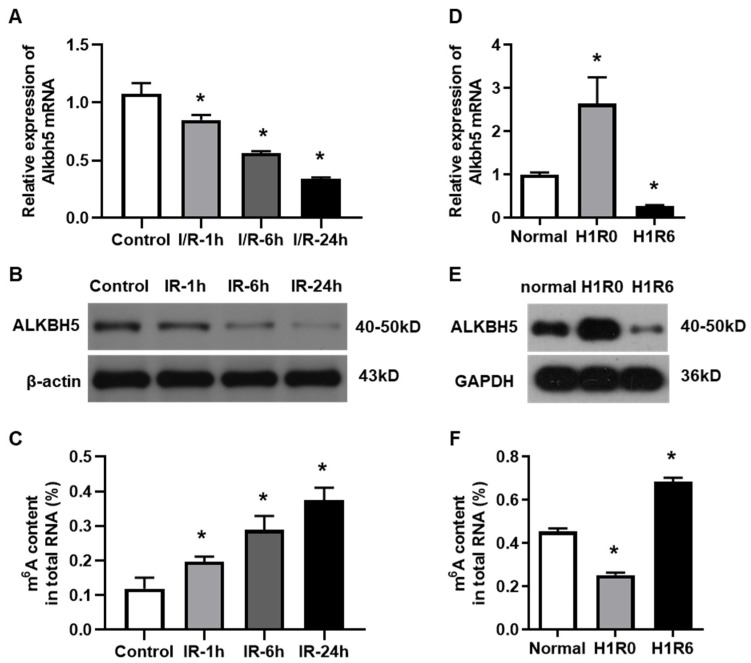
The expression of m^6^A demethylase ALKBH5 is decreased by hepatic I/R injury. (**A**,**B**) Alkbh5 mRNA (**A**) and protein (**B**) expression levels were detected with qPCR and western blot analysis in livers of mice subjected to I/R (ischemia 1 h and subsequent reperfusion for 1, 6, or 24 h). (**D**,**E**) Alkbh5 mRNA (**D**) and protein (**E**) expression levels in primary hepatocytes subjected to hypoxia or H/R (hypoxia 1 h and subsequent reoxygenation for 6 h). (**C**,**F**) Total levels of methylated mRNA (m^6^A) in hepatic tissues (**C**) or primary hepatocytes (**F**) after I/R or H/R. *n* = 6 mice for each group. * *p* < 0.05 compared to control or normal group.

**Figure 2 ijms-25-11537-f002:**
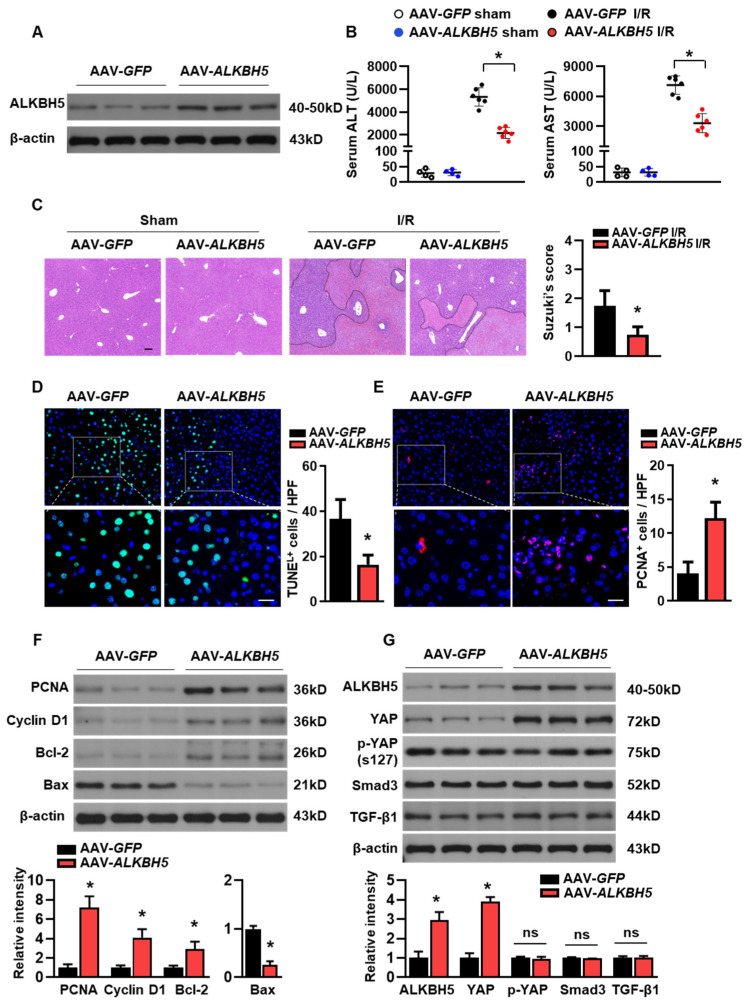
ALKBH5 overexpression ameliorates hepatic I/R injury. (**A**) ALKBH5 protein expression in livers from AAV-GFP and AAV-ALKBH5 mice was verified by western blot analysis. (**B**) Serum transaminases levels of AAV-GFP and AAV-ALKBH5 mice in the sham group or I/R (1 h/24 h) group. (**C**) Representative images with H&E staining and Suzuki score of liver sections from AAV-GFP and AAV-ALKBH5 mice subjected to sham or I/R treatment. The outlined areas in H&E staining images indicated hepatic necrosis. Scale bar, 100 μm. (**D**) Representative images with TUNEL staining and statistical analysis of positive cells in liver tissues from indicated mice (TUNEL-positive cells were stained with green). Scale bar, 20 μm. (**E**) Representative images with PCNA staining and statistical analysis of positive cells (PCNA-positive cells were stained with red). Scale bar, 20 μm. (**F**,**G**) Western blot determined expression patterns of indicated proteins in livers from indicated mice. *n* = 6 mice for each group. * *p* < 0.05 compared to the AAV-GFP control group. ns, no significance.

**Figure 3 ijms-25-11537-f003:**
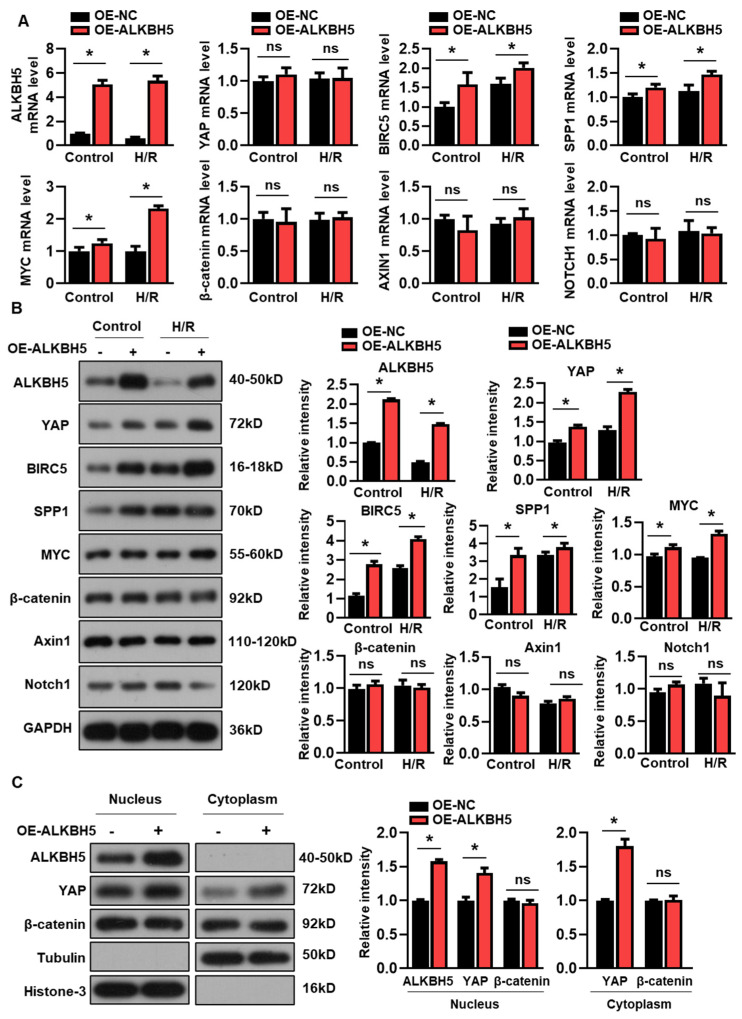
ALKBH5 promotes protein expression of YAP in hepatocytes subjected to H/R. (**A**) mRNA expression levels of indicated genes in H/R-treated L02 cells with ALKBH5 overexpression (OE). (**B**) Expression patterns of indicated proteins in L02 cells from indicated group. (**C**) Western blot determined expression patterns of indicated proteins in nucleus and cytoplasm of hepatocytes from the indicated group. The histogram shows the statistical analysis of the indicated protein expression. * *p* < 0.05 compared to negative control (NC) group. ns, no significance.

**Figure 4 ijms-25-11537-f004:**
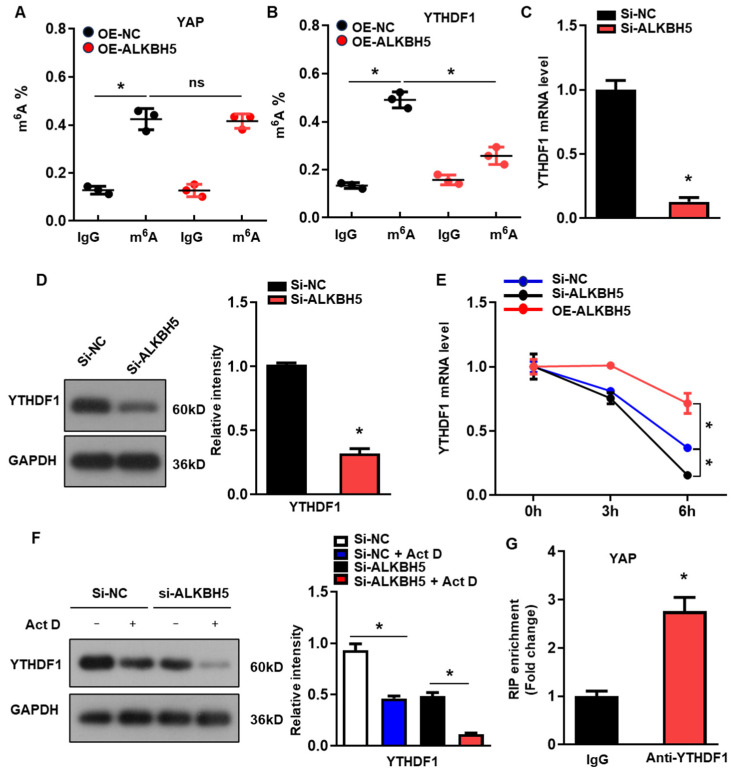
ALKBH5 regulates YTHDF1 expression via m^6^A demethylation to promote the translation of YAP. (**A**) m^6^A-meRIP-qPCR analysis of m^6^A level in YAP transcripts in L02 cells from indicated group. (**B**) m^6^A-meRIP-qPCR analysis of YTHDF1 in L02 cells from indicated group. (**C**) mRNA expression levels of YTHDF1 in cultured L02 cells from indicated group. (**D**) Protein expression of YTHDF1 in cultured L02 cells from indicated group. (**E**) qPCR analysis of YTHDF1 transcripts in Act D-treated L02 cells transfected with indicated expression plasmid. (**F**) Protein expression of YTHDF1 in Act D-treated L02 cells from indicated group. (**G**) RIP-qPCR analysis of the interaction of YAP and YTHDF1 in L02 cells. Enrichment of YAP was normalized to input. *n* = 3 independent experiments. * *p* < 0.05 compared to the corresponding control group. OE, overexpression; NC, negative control. Act D, actinomycin D.

**Figure 5 ijms-25-11537-f005:**
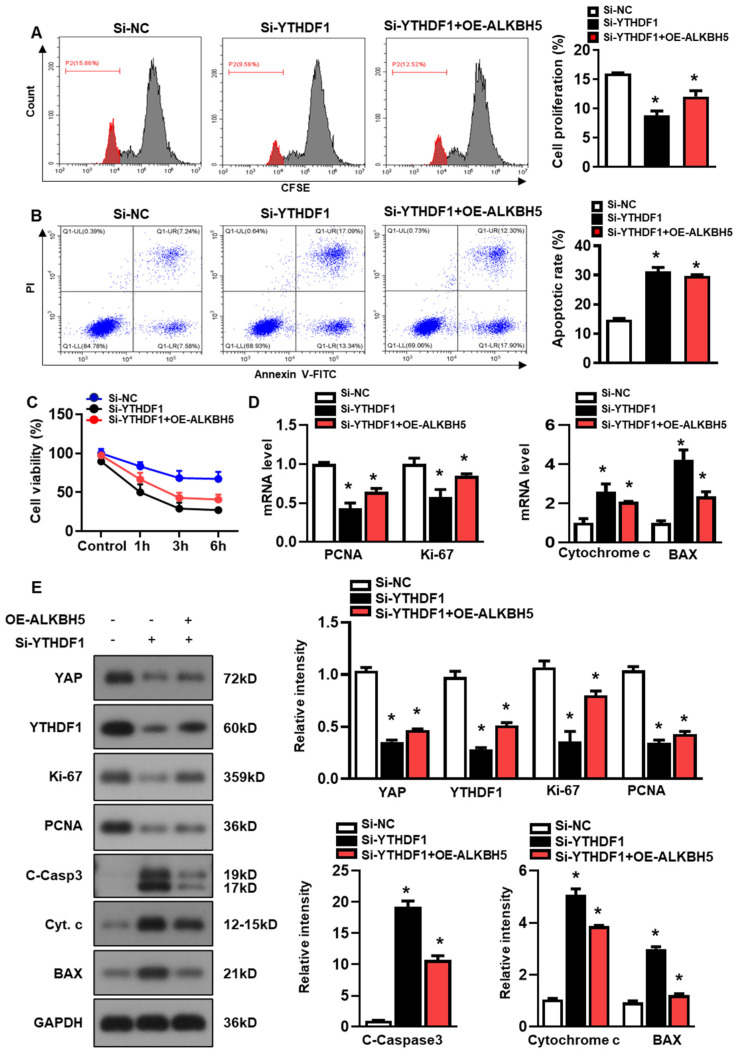
ALKBH5 promotes hepatocyte recovery from H/R-induced damage by regulating YTHDF1-YAP expression. (**A**) Cell proliferation in L02 cells with indicated plasmids after H/R treatment. (**B**) Flow cytometry analysis of cell apoptosis in indicated group. The histogram shows the percentage of total apoptotic cells, which was quantified as the sum of early apoptotic (annexin V-positive) and late apoptotic cells (annexin V-PI-double positive). (**C**) Cell viability in L02 cells from indicated group subjected to H/R for different time. *n* = 3 independent experiments. (**D**) mRNA expression levels of indicated genes in H/R-treated L02 cells with indicated plasmids. (**E**) Expression patterns of indicated proteins in H/R-treated L02 cells with indicated plasmids. c-Casp3, cleaved caspase-3; Cyt C, Cytochrome C. * *p* < 0.05 compared to the Si-NC control group.

**Figure 6 ijms-25-11537-f006:**
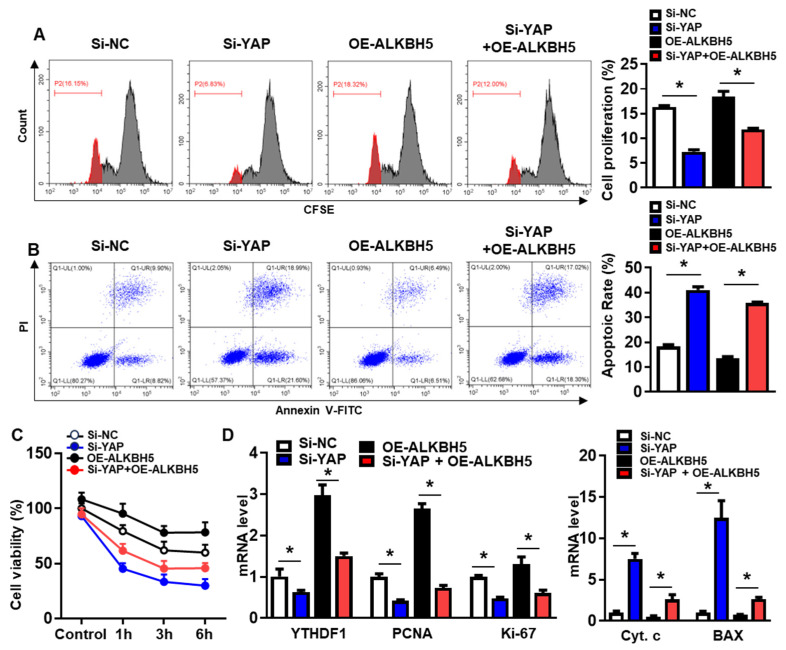
YAP expression is responsible for ALKBH5 function in hepatocyte activity in response to H/R damage. (**A**) Cell proliferation in L02 cells with indicated plasmids after H/R treatment. (**B**) Flow cytometry analysis of cell apoptosis in indicated group. The histogram shows the statistical analysis of total apoptotic cells from 3 identical experiments. (**C**) Cell viability in L02 cells from indicated group subjected to H/R for different time. *n* = 3 independent experiments. (**D**) mRNA expression levels of indicated genes in H/R-treated L02 cells with indicated plasmids. (**E**) Expression patterns of indicated proteins in H/R-treated L02 cells with indicated plasmids. c-Casp3, cleaved caspase-3; Cyt C, Cytochrome C. (**F**) Representative images with Ki-67 or BAX staining and statistical analysis of indicated protein expression in L02 cells from indicated group. Scale bar, 25 μm. * *p* < 0.05 compared to the corresponding control group. ns, no significance.

**Figure 7 ijms-25-11537-f007:**
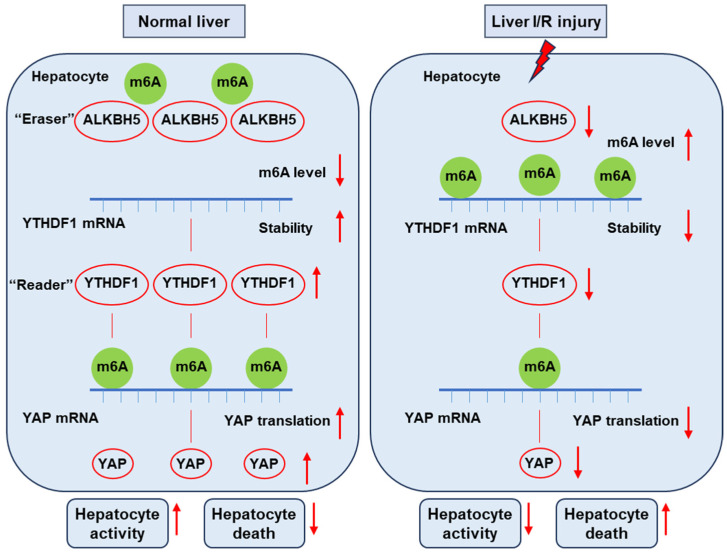
This schematic diagram depicts the proposed mechanisms of ALKBH5 in hepatic I/R injury. Under normal conditions, m^6^A “eraser” ALKBH5 downregulates the methylation of m^6^A “reader” YTHDF1 to promote the stability of YTHDF1 mRNA, and then YTHDF1 promotes the efficient translation of YAP. On the contrary, upon I/R stimuli, decreased ALKBH5 in the hepatocyte is accompanied with decreased YTHDF1 and low levels of YAP content, thus reducing hepatocyte activity and increasing cell death and eventually leading to severe liver damage.

## Data Availability

The data used to support the findings of this study are included within the article.

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
