# Peer review of "ALKBH5 Protects Against Hepatic Ischemia–Reperfusion Injury by Regulating YTHDF1-Mediated YAP Expression"

_ijms, 2024, doi:10.3390/ijms252111537_

Round 1

Reviewer 1 Report

Comments and Suggestions for Authors The manuscript is well written and contains no significant omissions. I have one general remark. For some important data in material and methods (e.g. animal groups, description of methods - important data related to a particular method etc...) the authors have stated: 'Detailed methods had been described in our previous study'. Unfortunately, I can not open reference 32 (doi.org/10.1038/nm.4290) and I can not see the study design... and that can be a problem for many authors. I have asked my colleagues from other institutions to download the publication and send it to me, therefore, I suggest you to at least provide the most important data (not in detail) in the current manuscript, even if they are already in the other study.

Introduction: In the second paragraph, please provide some information about the pathway you are investigating (YAP etc.), ...in the last paragraph is a one-sentence machine description...

Author Response

Response to Reviewer #1

Reviewer 1 Comments to the Author

The manuscript is well written and contains no significant omissions. I have one general remark. For some important data in material and methods (e.g. animal groups, description of methods - important data related to a particular method etc...) the authors have stated: 'Detailed methods had been described in our previous study'. Unfortunately, I can not open reference 32 (doi.org/10.1038/nm.4290) and I can not see the study design... and that can be a problem for many authors. I have asked my colleagues from other institutions to download the publication and send it to me, therefore, I suggest you to at least provide the most important data (not in detail) in the current manuscript, even if they are already in the other study.

Response: We appreciate the Reviewer’s conscientious in processing our manuscript. As suggested, we added detailed information in the revised Methods, including AAV expression vectors, animal groups and per group number of animals used. Besides, we have reorganized corresponding contents in the Materials and Methods part (highlighted in yellow).

Introduction: In the second paragraph, please provide some information about the pathway you are investigating (YAP etc.), ...in the last paragraph is a one-sentence machine description...

Response: We appreciate the Reviewer’s constructive suggestion. According to the Reviewer’s suggestion, we have reorganized corresponding contents and added more information and related references in the revised Introduction (highlighted in yellow).

Reviewer 2 Report

Comments and Suggestions for Authors

Dear Authors, 

The submitted manuscript is a very well prepare and would be of particular importance for the readers in the field.

Honestly, the image are very accurate, the statistical analysis is ok, the introduction and the discussion very well prepared.

You could also add possible related function of the ALKBH5 with other proteins or genes, using KEGG pathways analysis and STRING.

Also, did you find in your Experimental procedure any potential polymorphism in the sequence of the gene that could alter its function. 

Author Response

Response to Reviewer #2

Reviewer 2 Comments to the Author

The submitted manuscript is a very well prepare and would be of particular importance for the readers in the field.

Honestly, the image are very accurate, the statistical analysis is ok, the introduction and the discussion very well prepared.

  1. You could also add possible related function of the ALKBH5 with other proteins or genes, using KEGG pathways analysis and STRING.

Response: We thank the reviewer for highly evaluating our study. To our knowledge, this study is the first to reveal that ALKBH5 is a protective factor against hepatic I/R injury. Specifically, our results showed that ALKBH5-mediated m6A modification could improve hepatocyte repair and proliferation by maintaining YTHDF1 stability and YAP contents. On the basis of researching relevant literatures, the role of ALKBH5 and its related signaling pathway remain largely unknown, especially in the liver. Thus, it is difficult to search for potential target effectors that involves its function in liver I/R injury, using KEGG pathways analysis and STRING. However, we agree with the Reviewer that function of the ALKBH5 and its relationship with other proteins or genes needs further exploration.

  1. Also, did you find in your Experimental procedure any potential polymorphism in the sequence of the gene that could alter its function.

Response: Thank you for pointing this out. In general, several typical single nucleotide polymorphisms (SNP) in ALKBH5 (rs9913266, rs12936694) genes had been reported to affect the risk of some chronic diseases, such as cancer and immune-related disease (Adv Med Sci. 2021 Sep;66(2):351-358; Pharmgenomics Pers Med. 2022 May 31:15:547-559). However, liver ischemia/reperfusion injury is an acute injury. The relationship between the pathogenesis mechanism of liver I/R injury and the potential gene polymorphism remain largely unknown. To be honest, we didn’t perform any experiments involving ALKBH5 gene polymorphism in our study, which is out of our research field.

Reviewer 3 Report

Comments and Suggestions for Authors

The manuscript investigated the potential liver-protective action of ALKBH5 on hepatic ischemia-reperfusion injury and the underling mechanisms. They found that hepatocyte ALKBH5 is a protective factor against hepatic I/R injury by regulating YTHDF1-mediated YAP Expression. The study is well designed. I have some comments that should be addressed.

1. As transcription factors, the activity of YAP and β-catenin is mainly reflected by their nuclear migration. Thus, detecting their gene expression levels is of little significance, especially there is no any changes in the mRNA levels. The nuclear status of YAP and β-catenin (Cytoplasmic and nuclear levels of YAP and β-catenin) as well as phosphorylated YAP should be detected by WB. Especially, there is a significant change in the total YAP protein but not YAP mRNA, does it indicate that it affects the degradation/phosphorylation of YAP?

2. Regarding the cell proliferation and apoptosis, cell cycle analysis is necessary. Caspase-3 should be detected. Early and late apoptotic cells can be counted separately.

3. WB results need molecular size of each protein.

Author Response

Response to Reviewer #3

Reviewer 3 Comments to the Author

The manuscript investigated the potential liver-protective action of ALKBH5 on hepatic ischemia-reperfusion injury and the underling mechanisms. They found that hepatocyte ALKBH5 is a protective factor against hepatic I/R injury by regulating YTHDF1-mediated YAP Expression. The study is well designed. I have some comments that should be addressed.

  1. As transcription factors, the activity of YAP and β-catenin is mainly reflected by their nuclear migration. Thus, detecting their gene expression levels is of little significance, especially there is no any changes in the mRNA levels. The nuclear status of YAP and β-catenin (Cytoplasmic and nuclear levels of YAP and β-catenin) as well as phosphorylated YAP should be detected by WB. Especially, there is a significant change in the total YAP protein but not YAP mRNA, does it indicate that it affects the degradation/phosphorylation of YAP?

Response: We appreciate the Reviewer’s constructive suggestion. According to the Reviewer’s suggestion, we performed experiments to consolidate the conclusions and improve the quality of our study. Accordingly, the text, including the Materials and Methods, Results, and Figure Legends has been revised (highlighted in yellow).

Indeed, our earlier results showed that ALKBH5 overexpression increased the protein expression of YAP in liver tissues after I/R injury (Fig. 2F), while the phosphorylation of YAP at S127 (p-YAP) were not affected by ALKBH5 overexpression (revised Fig. 2F). Besides, our further results showed that ALKBH5 protein was only detectable in the nuclear fractions, enabling m6A modification of pre-mRNA, which was consistent with previous study (Mol Cancer. 2020 Feb 27;19(1):40) (Fig. 3C). Of note, ALKBH5 overexpression increased the YAP protein contents in both nucleus and cytoplasm (Fig. 3C). By contrast, ALKBH5 overexpression did not affect the protein expression of β-catenin, neither in nucleus nor in cytoplasm (Fig. 3C). Taken together, our results indicated that ALKBH5 promotes the protein expression of YAP through YTHDF1-mediated YAP translation, which was not related to degradation or phosphorylation of YAP.

  1. Regarding the cell proliferation and apoptosis, cell cycle analysis is necessary. Caspase-3 should be detected. Early and late apoptotic cells can be counted separately.

Response: Thank you for pointing this out. To reveal the role of ALKBH5-m6A-YTHDF1-YAP axis in regulating hepatocyte proliferation, the expression of cell proliferation related markers (PCNA and Ki-67) was assayed (Fig. 5D and 5E, Fig. 6D and 6E). Ki-67 and proliferating cell nuclear antigen (PCNA) are commonly used to identify proliferating cells from resting cells (Curr Protoc Mol Biol. 2015 Jul 1:111:28.6.1-28.6.11). To be specific, Ki-67 is highly expressed in proliferating cells (maximally in G2 and early M phase), is rapidly degraded during anaphase and telophase, and is rarely expressed in the G0 phase (J Immunol. 1984 Oct;133(4):1710-5). PCNA marks proliferating cells and is most highly expressed in S-phase cells (Int J Cancer. 1983 Jan 15;31(1):13-20). We agree with the reviewer, to further dissect cell cycle phases need more detailed experiments, which is out of the field of this study and could be further researched in next study. Based on the Reviewer’s suggestion, we added above contents and related references in the revised Results.

Indeed, to reveal the influences and results of cell apoptosis, western blotting analysis for cleaved caspase-3 (c-Casp3) were included in revised Fig.5E and Fig.6E.

Flow cytometry analysis in Fig. 5B and Fig. 6B showed the results of cell apoptosis, including early apoptotic (annexin V-positive) and late apoptotic cells (annexin V-PI-double positive). To make the results display more concise, the percentage of total apoptotic cells was quantified as the sum of early apoptotic and late apoptotic cells.

  1. WB results need molecular size of each protein.

Response: According to the Review’s suggestion, we have added the kd value for all Western blots.